# Transport Cost Estimation Model of the Agroforestry Biomass in a Small-Scale Energy Chain

Giulio Sperandio [1,*] , Andrea Acampora [1] , Vincenzo Civitarese [1] , Sofia Bajocco [2] and Marco Bascietto [1]

1   Consiglio per la ricerca in agricoltura e l'analisi dell'economia agraria (CREA)—Centro di ricerca Ingegneria e Trasformazioni agroalimentari, Via della Pascolare 16, 00015 Roma, Italy; andrea.acampora@crea.gov.it (A.A.); vincenzo.civitarese@crea.gov.it (V.C.); marco.bascietto@crea.gov.it (M.B.)
2   Consiglio per la ricerca in agricoltura e l'analisi dell'economia agraria (CREA)—Centro di ricerca Agricoltura e Ambiente, Via della Navicella 4, 00184 Roma, Italy; sofia.bajocco@crea.gov.it
*   Correspondence: giulio.sperandio@crea.gov.it

**Abstract:** The delivery of biomass products from the production place to the point of final transformation is of fundamental importance within the constitution of energy chains based on biomass use as a renewable energy source. Transport can be one of the most economically expensive operations of the entire biomass energy production process, which limits choices in this sector, often inhibiting any expansive trends. A geographic identification, through remote sensing and photo-interpretation, of the different biomass sources was used to estimate the potential available biomass for energy in a small-scale supply chain. This study reports on the sustainability of transport costs calculated for different types of biomass sources available close a biomass power plant of a small-scale energy supply chain, located in central Italy. To calculate the transport cost referred to the identified areas we used the maximum travel time parameter. The proposed analysis allows us to highlight and visualize on the map the areas of the territory characterized by greater economic sustainability in terms of lower transport costs of residual agroforestry biomass from the collection point to the final point identified with the biomass power plant. The higher transport cost was around €40 Mg$^{-1}$, compared to the lowest of €12 Mg$^{-1}$.

**Keywords:** energy chain; residual biomass; isochronous rings; travel time; transport cost





## 1. Introduction

In recent years, the European Union has devoted growing attention to supporting and promoting actions to combat climate change that, together with the policies aimed at achieving greater energy efficiency, can foster development of a more sustainable economic-energy and environmental system [1]. All this would result in less dependence on imports, less environmental impact, and an increase in added value for rural areas. The economic and social interest for renewable energies and bioenergy can be attributed to various aspects including, mainly, the possibility of reaching energy self-supply levels more easily and, at the same time, heavily replacing the use of fossil fuels [2–4]. In this way it is possible to obtain the double effect of reducing the emissions of greenhouse gases into the atmosphere and of respecting the commitments made at world level with the ratification of various international treaties starting from the Kyoto Protocol of 1997, favoring the possibility of increasing the added value and the employment level of the local energy supply chains [5,6]. There are different types of biomass that can be used for energy purposes, both in raw form as firewood, and as woodchip or as refined material such as pellets and briquettes [7–9], which have favored the development of technologies suitable for the optimal conversion of these products into energy (heat and electricity). This product diversity corresponds to the possible activation of equally diversified energy chains to be organized in rural energy districts. The implementation of a bioenergy supply chain is mainly linked to its efficient logistics organization [10], on which most of the impacts (economic, occupational,

environmental, etc.) depend, and the real convenience to the production of primary energy from agroforestry biomass. It is therefore essential to make the entire energy transformation production process sustainable when referring to the different options available, about the aspects relating to the collection, transport, and energy transformation, applying models that can improve the efficiency of the production processes and optimizing production costs for every available biomass typology [11–17].

The future challenges are aimed at improving the competitiveness of the biomass energy chain and all go in the direction of sustainable development, which involves not only the technological and production aspects, but also the governance and organization of the sector [14,18]. These aspects must inevitably be correlated to what is one of the key factors that regulate, influencing its efficiency, the entire energy transformation system, represented by the cost of transporting biomass [19]. The transport of woody biomass, in fact, significantly affects the costs, the emissions of dust and the carbon balance within the energy chain [20], with increasingly negative impacts as a function of the increase in the distance traveled and the volume of biomass transported [21,22].

The logistics of the short energy supply chain and the priority valorization of the biomass spread throughout the territory represent the base for a sustainable development of bioenergy, which can be supported by the realization of small-medium power plants, compatible with the energy demands of users in the specific area of reference [23,24].

A small territorial basin, where different sources of agroforestry biomass are available, was taken as a case study. A geographic model was built to evaluate the cost-effectiveness of the logistics of agroforestry biomass transport, in the context of the small local energy supply chain [25], taking into consideration both technical-economic data (transport, loading and unloading times and costs biomass), and data relating to the territory, such as type of biomass and its location, qualitative and quantitative availability, viability, and possible routes with respect to the transformation point.

## 2. Materials and Methods

### 2.1. Study Area and Biomass Power Plant

The study area refers to a small-scale supply chain located North-East of Rome and surrounding a biomass power plant, installed within the farm of CREA-Research Centre for engineering and agri-food processing in Monterotondo, Italy (42°6′2.63″ N; 12°37′37.36″ E).

The identification of the boundaries of the potential wooden biomass supply area was based on the travel time of the trucks from any spatial point located no more than 60 min away from the biomass plant, selecting the road with shortest travel time, excluding the highways [25]. For this reason, the shape of the study area is irregular and depends on the spatial arrangement of the road network, road types and speed limits. Wider and linear roads, on which traffic can flow faster, make the study area wider than the narrower and more winding roads, on which, instead, the vehicle travels slower, reducing the study area. Five areas consisting of irregular isochronous rings were identified. The external boundaries of each ring were determined in relation to five-time bands depending on the time taken to travel from the specific biomass location to the plant. The first ring, outermost, with a travel time varying from 60 to 50 min, the second from 50 to 40 min, the third from 40 to 30 min, the fourth from 30 to 20 min and finally, the fifth ring, the closest to the biomass plant, from 20 to 0 min. The isochronous rings were calculated by applying the software R osrm package (OpenStreetMap-Based Routing Service) version 3.2.0 [26]. The furthest point of the external boundary of the area from the plant was at a linear distance of about 35 km, while the closest was about 16 km. The total area examined was 2276 km$^2$, distributed on each single isochronous ring in an increasing way starting from the fifth to the first.

The proposed model was constructed with reference to the specific point of energy transformation, represented by the biomass power plant used for the heating of the research Centre buildings, which are characterized by a potential volume to be heated of about 10,000 m$^3$. The biomass plant, equipped with a mobile grate, has a nominal heat output of

350 kW, but is also set up for micro-cogeneration by means of a steam turbine capable of generating 500 kg of steam per hour at a pressure of 12 bar, for a potential production of about 40 kWh of electricity. Currently the plant can only be used to produce thermal energy as the steam turbine is still being installed. For the building heating only, on a period of use of 130 days per year, the potential annually consumable biomass, with a water content of 35%, is around 290.1 Mg per year. When the micro-cogeneration option will be activated, then the plant could be used 24 h per day for about 300 days per year, with an annual biomass consumption estimated at 811.5 Mg.

### 2.2. Biomass Estimation

The potential annual supply of residual biomass from the observed area was estimated on a sampling considering the land cover area of each sampling population visualized by satellite images in Google Earth software [25]. Once the different types of biomass present in the area were identified, their quantitative estimation was subsequently carried out by applying judgment coefficients of experts of photo-interpretation. On a total of 139 observations, eight sampling populations were defined, each of which was independently sampled. In Table 1, the potential biomass range for each class and other evaluation parameters considered in the application of the biomass transport cost assessment model are reported.

**Table 1.** Parameters used to estimate the residual biomass potentially available from the different biomass sources.

| Biomass Classes | Density | Residual Biomass Production | Frequency of Intervention | Residual Biomass Per Tree |
|---|---|---|---|---|
| | Tree ha$^{-1}$ | Mg ha$^{-1}$ | Years | Mg Tree$^{-1}$ year$^{-1}$ |
| 1. Green urban area (GUA) | 80 | 16.0–32.0 | 8 | 0.0250–0.0500 |
| 2. Sport and leisure facilities (SLF) | 50 | 10.4–20.0 | 8 | 0.0250–0.0500 |
| 3. Vineyards (VIY) | 3000–4000 | 2.1–3.0 | 1 | 0.0007–0.0010 |
| 4. Fruit trees and berry plantations (FBP) | 400–500 | 2.0–3.5 | 1 | 0.0050–0.0070 |
| 5. Olive groves (OGR) | 180–300 | 3.6–8.1 | 2 | 0.0200–0.0270 |
| 6. Complex cultivation patterns (CCP) | 130–260 | 2.0–4.0 | 2 | 0.0100–0.0135 |
| 7. Land principally occupied by agriculture (LOA) | 400–500 | 2.0–3.5 | 1 | 0.0050–0.0070 |
| 8. Forests (FOR) | 800–1000 | 18.8–26.3 | 25 | 0.0009–0.0011 |

Based on the coverage of the tree canopy observed on the territorial map, four levels of biomass production (in Mg ha$^{-1}$ year$^{-1}$) have been attributed for each of the eight types of biomass. L3, L2, L1 and L0 are the classes considered in the study and indicate, respectively, the maximum, average, minimum or absent value of the available biomass. Table 2 shows the values of these parameters.

**Table 2.** Biomass production levels (L) for the calculation of residual biomass available for each typological class (in Mg ha$^{-1}$ year$^{-1}$).

| Typology | L3 | L2 | L1 | L0 |
|---|---|---|---|---|
| 1. Green Urban Areas (GUA) | 4.00 | 3.00 | 2.00 | 0.00 |
| 2. Sport and Leisure Facilities (SLF) | 2.50 | 1.90 | 1.30 | 0.00 |
| 3. Vineyards (VIY) | 3.00 | 2.55 | 2.10 | 0.00 |
| 4. Fruit Trees and berry Plantation (FTP) | 3.50 | 2.75 | 2.00 | 0.00 |
| 5. Olive Groves (OGR) | 4.00 | 2.90 | 1.80 | 0.00 |
| 6. Complex Cultivation Patterns (CCP) | 2.00 | 1.50 | 1.00 | 0.00 |
| 7. Land principally Occupied by Agriculture (LOA) | 3.50 | 2.75 | 2.00 | 0.00 |
| 8. Forest class (FOR) | 1.05 | 0.90 | 0.75 | 0.00 |

In this way, a set of data referring to each single areas of the map was constructed, identifying the type, the overall surface, the percentage of surface referring to the different biomass level potentially available, and that belonging to a specific isochronous ring.

The biomass potentially available in each sample area was then determined on the basis of the biomass estimated for each sub-area relating to each production level for each type of biomass class. The production level was determined in relation to the density of the respective typological classes. Figure 1 shows an example representation of the procedure for estimating the areas and biomass potentially available for each typological class. The biomass for each circular sample area was then determined by applying the following formula (Equation (1)):

$$Biomass\ (\text{Mg})\ =\ \sum_{k=0}^{3}\left(A_{kj}\ \times\ L_k\right) \tag{1}$$

where:

1. $A$—area in hectares corresponding to the level L of biomass production (Table 1);
2. $k$—number of areas corresponding to the same production level within the sample area; and
3. $j$—number of areas belonging to the same production level $L_k$.

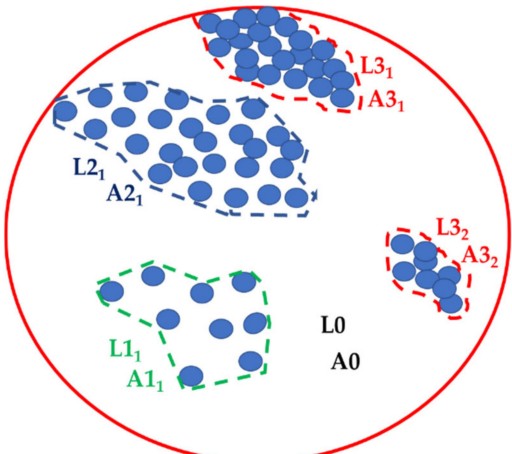

**Figure 1.** Graphic representation of the estimated areas ($A_{kj}$) corresponding to the biomass production levels ($L_k$) within a single sample area, calculated in relation to the vegetation density.

### 2.3. Transport Cost Evaluation Model

The analysis of the economic sustainability of the recovery of the biomass available on the territory of the small supply chain was based on the evaluation of the costs of the biomass transport operation (round trip of the truck) from the loading site to the biomass plant, including loading and unloading operations. For the acquisition of the biomass, the hypothesis adopted is that farmers provide free pruning biomass to avoid incurring the tariffs applied for the disposal of this material in landfills. Both the farmer and the manager of the plant benefit from the agreement—the farmer does not pay for the disposal of biomass residues; the latter does not pay for the recovered raw material.

In the case study, the pruning biomass should already be staked by the farmer and directly available for loading and transport (Figure 2a). An unloaded truck is assumed to leave the biomass plant and arrive at the biomass loading site, where a forest loader loads the stacked biomass (Figure 2b).

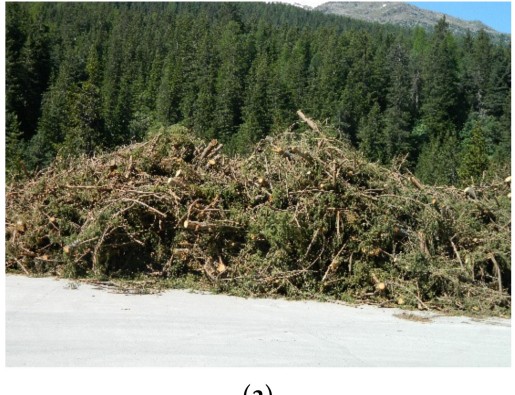
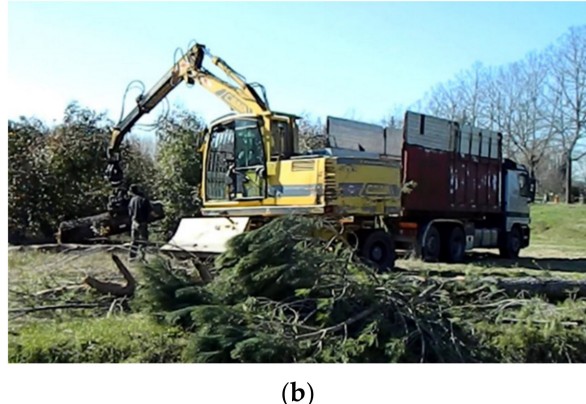

(**a**)                                                                (**b**)

**Figure 2.** Pruning residues stacked (**a**) and biomass loading operation on the truck (**b**).

Once the loading operation is complete, the truck travels to its destination and unloads the biomass near the plant. The truck carries out subsequent travels until the schedule daily work time is completed, considering about 8 working hours per day. Given the small size of the processing plant, it was assumed that this operation was carried out using a truck with a total load capacity of 26 m$^3$, corresponding to an estimated residual biomass weight of about 8 Mg. A forest loader equipped with a grapple to carry out the loading/unloading of biomass must be daily transferred to the workplace and brought back using a dedicated truck. The hourly costs of the machines calculated with analytical methodology [27], and the principal economic and technical elements considered, are reported in Table 3.

**Table 3.** Main elements considered for the calculation of the hourly cost of machines and labor.

|  | Truck for Biomass Transport | Truck for Loader Transport | Forest Loader |
|---|---|---|---|
| Purchase price (€) | 110,000 | 95,000 | 80,000 |
| Salvage value (€) | 7559 | 6528 | 8590 |
| Life time (years) | 12 | 12 | 10 |
| Total time (h) | 14,400 | 14,400 | 8000 |
| Engine Power (kW) | 309 | 280 | 88 |
| Interest rate (%) | 4.0 | 4.0 | 4.0 |
| Fuel consumption (L h$^{-1}$) | 25.49 | 23.10 | 9.44 |
| Fuel price (€ L$^{-1}$) | 1.50 | 1.50 | 1.10 |
| Driver cost (€ h$^{-1}$) | 21.00 | 21.00 | 15.00 |
| Machine cost (€ h$^{-1}$) | 71.00 | 64.00 | 35.00 |
| Total operating cost (€ h$^{-1}$) | 92.00 | 85.00 | 50.00 |

The formula used to determine the unitary transport cost (Equation (2)), including the biomass loading, transport and unloading cost and the daily forest loader transfer cost, is the following:

$$BTC = \frac{[(Ttr \ \times \ Ctr] + (Tlu \ \times \ Clo) + (tc \ \times \ Ctl))}{bl} \quad (2)$$

where:

1. *BTC*—biomass transport cost per Mg (€ Mg$^{-1}$);
2. *Ttr*—roundtrip travel time, obtained doubling the return travel time of the loaded truck (h);
3. *Tlu*—time required for loading and unloading operations (h);
4. *Ctr*—hourly cost of the truck (€ h$^{-1}$);
5. *Clo*—hourly cost of the loader (€ h$^{-1}$);
6. *tc*—loader transferring coefficient;

7. *Ctl*—hourly cost of the truck that transfer the loader to destination and return (€ h$^{-1}$); and
8. *bl*—average biomass load considered per travel (Mg).

The travel time (*Ttr*) was calculated by doubling the average time attributed to each of the five isochronous rings (h). An average time of 75 min was considered as the basis for calculating the *Tlu*, divided into about 56 min (75%) for loading and 19 min (25%) for unloading. To consider the influence of the different types and quantities of biomass on the loading phase, two multiplying coefficients were considered. *Tlu* was then calculated according to the following Equation (3).

$$Tlu = (Tl \times lc) \cdot (1 + yc) + Tu \tag{3}$$

where:

1. *Tl*—loading time (h);
2. *Tu*—unloading time (h)
3. *lc*—load coefficient depending on biomass type; and
4. *yc*—yield coefficient depending on the quantity of biomass per hectare.

The impact of the loader transfer on the total transport cost per Mg was estimated considering a loader transfer coefficient (*tc*) for each biomass class. In fact, the daily transfer time of the loader (one round trip) must be added to the total daily time used for transporting the biomass. The greater the number of biomass transport trips per day, the lower this coefficient, since the latter is calculated as the reciprocal of the number of daily trips. Table 4 shows the average values of the correction coefficients *lc*, *yc* and *tc*.

**Table 4.** Average coefficients used for the calculation of final travel time (*lc*, load coefficient; *yc*, yield coefficient; *tc*, loader transfer coefficient).

| Typology | Coefficients | | |
|:---:|:---:|:---:|:---:|
| | *lc* | *yc* | *Tc* |
| GUA | 1.00 | 0.20 | 0.37 |
| SLF | 1.05 | 0.29 | 0.34 |
| VIY | 1.15 | 0.21 | 0.43 |
| FTP | 1.05 | 0.21 | 0.33 |
| OGR | 1.10 | 0.23 | 0.34 |
| CCP | 1.10 | 0.30 | 0.35 |
| LOA | 1.15 | 0.21 | 0.34 |
| FOR | 1.00 | 0.27 | 0.30 |

To assess the economic sustainability of the recovery and transport of biomass scattered over the territory of the small-scale energy chain, it was necessary to consider that the biomass unloaded at the plant had to be chipped before use. A positive judgment on economic sustainability was based on the positive difference between the average market value of the woodchip and the cost incurred for transport and chipping. For the chipping operation, a forest chipper available on the farm was considered. The average cost of this operation was estimated at 15 € Mg$^{-1}$. The market value of woodchip from biomass residues was instead estimated at around 45 € Mg$^{-1}$.

## 3. Results

The results are expressed in relation to the different typologies of biomass sources. Figure 3 shows the estimate of the quantity of biomass available in the territory examined in relation to the production levels (L) and the biomass classes. The land principally occupied by agriculture (LOA) class is the one with the highest average value (2.13 Mg ha$^{-1}$ year$^{-1}$), with the highest incidence of L3 (3.07 Mg ha$^{-1}$ year$^{-1}$). The class that contributes less is SLF with 0.25 Mg ha$^{-1}$ year$^{-1}$. In Figure 4, we report the average time consumed (Figure 4a) and relative average costs (Figure 4b) for the load, transport, and unload operations of

the residual biomass for each class. The highest time is request for the VIY class with 4.23 h trip$^{-1}$, while the shortest time is recorded for the FOR class, with 3.04 h trip$^{-1}$. The other biomass classes record intermediate times between 3.05 and 3.50 h trip$^{-1}$. The load/unload time is highest in the CCP class with 1.65 h, followed by LOA and VIY with 1.61 h, while GUA requires the lowest time of 1.44 h. The trend in average costs per trip reflects that the times with the highest value of €316.31 trip$^{-1}$ was VIY, corresponding to €39.54 Mg$^{-1}$, and the lowest value was €213.84 trip$^{-1}$ for FOR, that is €26.73 Mg$^{-1}$.

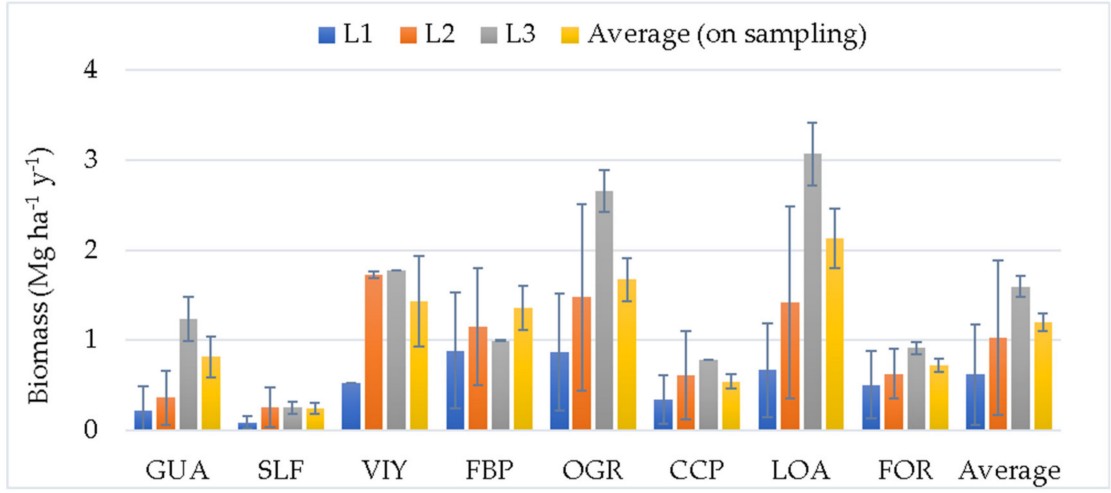

**Figure 3.** Average value (±standard error) of biomass available (Mg ha$^{-1}$ year$^{-1}$) in reference to the production level (L) and biomass class.

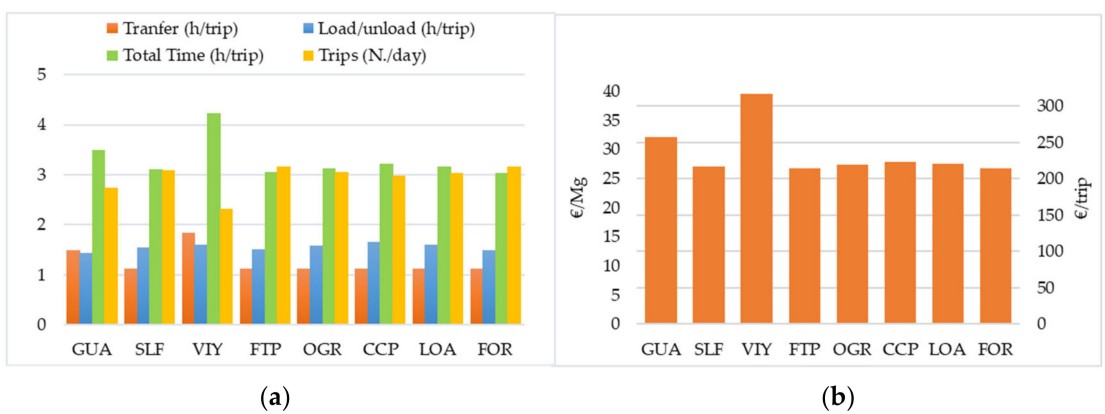

(**a**)                        (**b**)

**Figure 4.** Average time consumption (**a**) and costs (**b**) for the recovery and transport of residual biomass per class.

It should be noted that, with regard to the VIY class, it is not widespread in the observed area, therefore it is not to be considered a representative figure for this class. For the other classes, on the other hand, there is a greater homogeneity of results for the classes considered in the agricultural field such as FTP, OGR, CCP and LOA. Figure 5 shows the matrix graphs of the average transport cost (Figure 5a) and economic sustainability (Figure 5b) per Mg, in relation to the biomass classes and isochronous rings. In this figure, it is clearly seen how the distance from the biomass plant is very important. The cost increases by proceeding from the 5th isochronous ring (travel time 0–20 min) to the 1st (50–60 min).

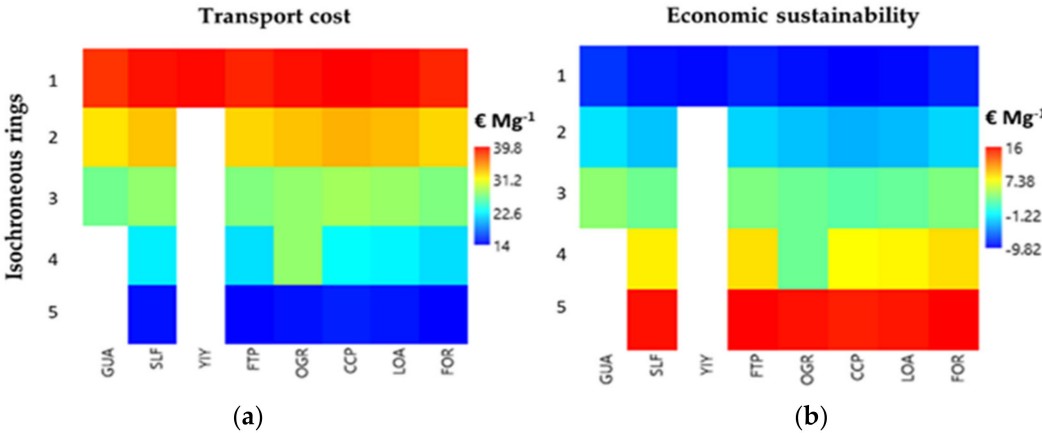

**Figure 5.** Matrix plots of the biomass transport cost (**a**) and economic sustainability of the transport operation (**b**), in relation to biomass classes and isochronous rings.

This is valid for all classes even if in a different way. The average costs vary from a minimum of about €14 Mg$^{-1}$ in the area of the 5th ring, to a maximum of about €39.80 Mg$^{-1}$ in the 1st ring. Economic sustainability is inversely proportional to the costs incurred. The red and yellow colors indicate the areas of greater sustainability, coincident with the 5th and 4th ring (positive values). The 3rd ring represents the boundary area between positive and negative values. The 2nd and 1st ring areas are always associated with negative values and indicate an uneconomic condition of the operation. Figure 6 shows the territorial map in which transport costs are associated with specific areas of the map. From this map it is possible to check the cost of transport (Figure 6a) in relation to the distance from the biomass loading point to the unloading point near the transformation plant. The conditions of economic sustainability (Figure 6b) show positive values in the areas ranging from yellow to blue (near the biomass plant).

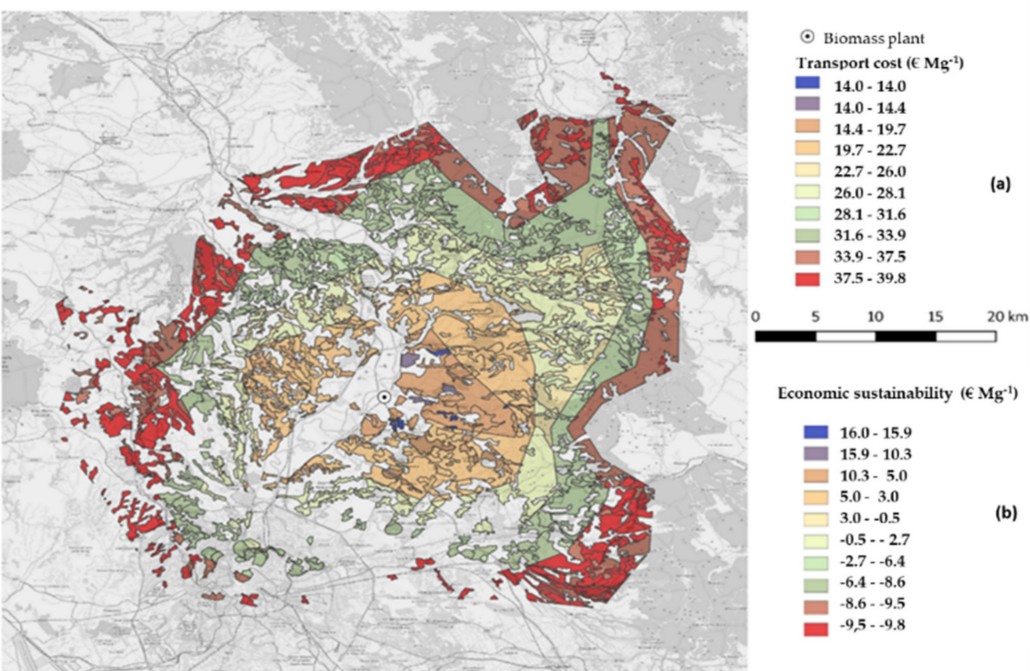

**Figure 6.** Territorial Map of the biomass transport cost (**a**) and economic sustainability of the operation (**b**) according to typological classes and isochronous rings.

In Figure 7, the total area (Figure 7a) and the related potential biomass available (Figure 7b) in the investigated area was compared with the area and quantity of biomass of greatest interest for the purposes of the study, where economic sustainability is verified, corresponding to the 4th and 5th isochronous rings.

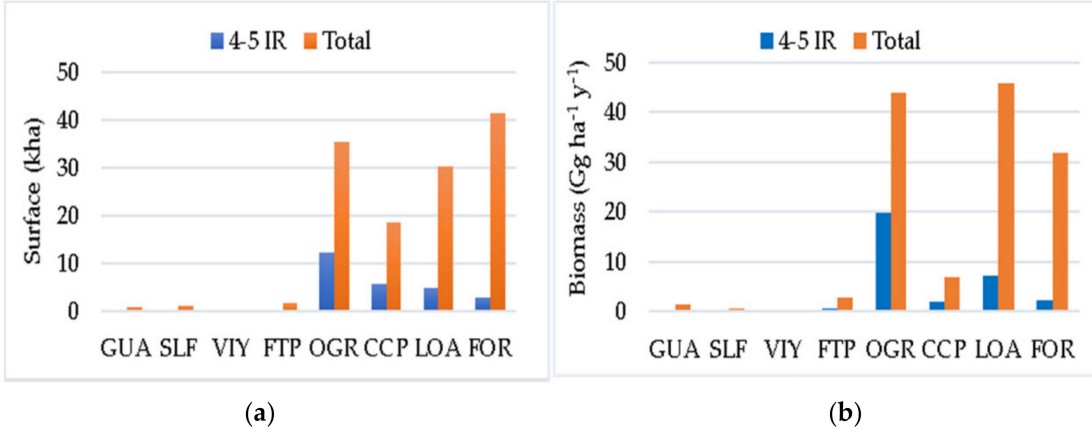

(**a**)                                    (**b**)

**Figure 7.** Total surface investigated and surface of the 4th and 5th isochronous ring (IR) (**a**) and total residual biomass available and biomass on 4th and 5th isochronous ring (IR) (**b**), where economic sustainability is verified, for each class considered.

The market price of the woodchip and the chipping cost can represent a critical aspect of the model. A sensitivity analysis on these parameters showed in Figure 8 reveals, for example, that a 20% reduction in the value of the woodchip (Figure 8a) would make most of the biomass classes uneconomic even in the ring closest to the biomass plant, while the same negative effect is obtained if the cost of the chipping operation increased by over 30% (Figure 8b). On the contrary, with the increase of the same percentage of the price, the recovery of biomass becomes convenient for almost all classes in all rings, while the same result is obtained if the cost of the chipping operation is reduced by more than 30%.

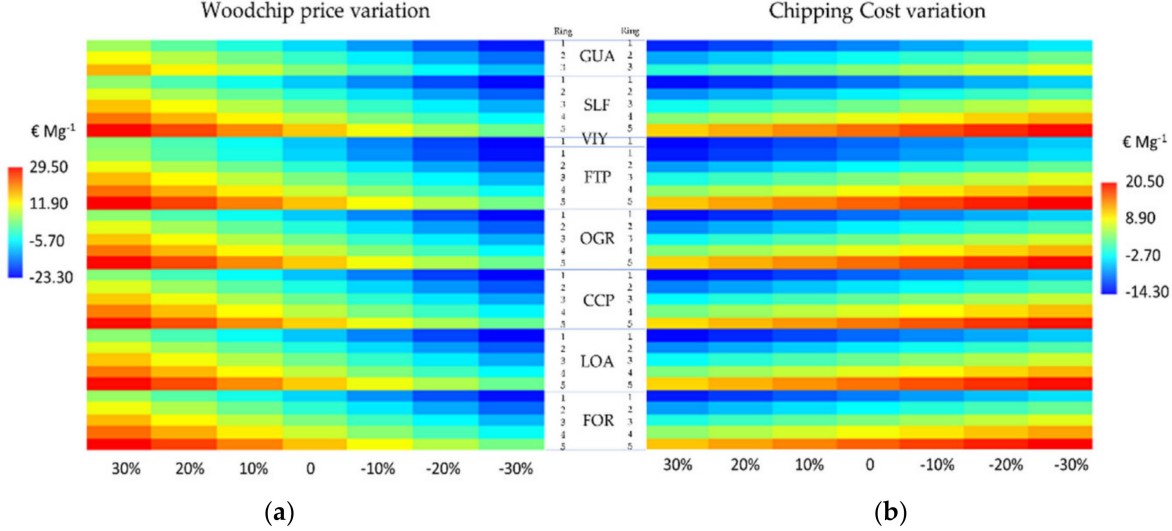

(**a**)                                    (**b**)

**Figure 8.** Sensitivity analysis related to economic sustainability (difference between woodchip price and recovery and chipping costs) in relation to the variation of woodchip price (**a**) and chipping cost (**b**), according to biomass classes and isochronous rings.

## 4. Discussion

The study has proposed a model to evaluate the cost and the relative economic sustainability of the recovery and transport of the potentially available residual biomass on a small-scale energy supply chain. The model was based on the cost evaluation of recovery of different potential biomass sources spread over the territory close to a small biomass plant. On a total observed area of 2276 km$^2$, about 57%, equal to 130 kha, was considered useful for the proposed model.

The annual residual biomass potentially available was about 134 Gg distributed on the territory observed. Much of the biomass available was classified within the agricultural area (LOA) for about 34.4%, followed by olive groves with 33.0% and forest area with 23.8%. The remaining 8.8% is mainly divided between associated crops and orchards. Of this total biomass available, only 24.0% falls within the area of the economic sustainability (5th and 4th isochronous ring) for a quantity of about 32 Gg, 62% of which is represented by olive grove pruning residues. Areas with agricultural crops and forests follow with an incidence of 23% and 7%, respectively. For the case study, only a small percentage of this available biomass can be used in the biomass plant, whose annual consumption in a cogeneration system does not exceed 1 Gg of woodchip. However, it is useful to have a tool to plan any biomass supplies under convenient conditions. The economic evaluation model is applied to the different types of biomass available, considering the various difficulties related to the quality of biomass residues and the influence this generates in the calculation of recovery and transport costs [28,29].

The cost of transporting biomass from the origin to its destination represents a significant share of the overall costs of supplying woody biomass in an energy chain. Some studies attribute to this cost a percentage between 25% and 50% of the total cost of the delivered biomass, depending on the transport distance, the bulk density of the load and the moisture content of the biomass transported [30,31]. Increasing the efficiency of transport operations should therefore lead to a significant reduction in related costs and with it also a probable reduction in environmental impacts.

The model, as expected, returns the highest cost values, for all biomass classes, correlating to the greatest distances to travel. For the examined small-scale energy chain, the economic sustainability for the supply of biomass to feed the power plant is verified when the travel distance does not exceed 20 km, with a travel time from the place where the biomass is loaded to the plant, no more than 35 min. At the same distance, the pruning residues of orchards and forest biomass are slightly more advantageous than the other classes, while the pruning of vineyards, being represented only within the most distant isochronous ring, is always uneconomical.

The economic sustainability in the model is calculated, for each class, considering the difference between the market price of the woodchip and the biomass transport and chipping costs. This last operation is very critical, and the average cost considered in this study could significantly increase in the case of agricultural pruning waste compared, for example, to forest residual biomass, reducing the economic convenience range [32].

The sensitivity analysis applied to these two parameters highlighted a greater sensitivity of the model to a change in the price of the woodchip compared to the same percentage change (of opposite sign) in the cost of the chipping operation. To maintain the same level of economic sustainability returned by the proposed model, a possible increase/decrease in the market value of the woodchip (decrease/increase in the cost of the woodchip) would correspond to an increase/decrease in the transport distance and, consequently, in the area affected by the recovery of the biomass.

## 5. Conclusions

The study carried out was aimed at implementing a geographic evaluation model capable of providing a mapping of the costs of transporting biomass (including loading and unloading) from production sites to processing sites. By mapping the cost of biomass transport, it is possible to orient the choices in relation to the size of the energy transforma-

tion plants to be considered also in a project to enhance the local resources available. The small-scale energy supply chain, in fact, currently represents a model to be encouraged and applied in farms that want to make a qualitative leap towards a bioenergy company. This model can be implemented on larger areas and on different territorial conditions and can be a useful knowledge base in the management of the bioenergy resources of a territory. In these cases, however, it is necessary to verify from time to time the boundaries of the potential biomass supply area, the road network, the type of biomass resources available and the level of productivity obtained in the investigated area to calibrate the coefficients of the economic model, on which the calculation of the final cost of transporting biomass depends.

Finally, a very interesting aspect of the proposed model concerns the possibility of starting a virtuous process of mutual benefit between the farmers of a territory and the bioenergy company, which translates into a recovery of residual biomass, otherwise destined for landfill, or burned in field. In this way, environmental impacts are also reduced thanks to a more controlled combustion process in small biomass plants. The model of the small-scale energy supply chain can represent the most suitable solution for the development of sustainable systems compatible with the availability of bioenergy that the territory is able to supply.

**Author Contributions:** Conceptualization, G.S.; methodology, G.S., M.B. and S.B.; formal analysis, G.S.; data curation, G.S.; writing—original draft preparation, G.S.; writing—review and editing, G.S., A.A., V.C., M.B. and S.B. All authors have read and agreed to the published version of the manuscript.

**Funding:** This research was funded by Italian Ministry of Agriculture, Food and Forestry Policies (MiPAAF), grant D.D. n. 26329, 1 April 2016, project AGROENER "Energia dall'agricoltura: innovazioni sostenibili per la bioeconomia".

**Acknowledgments:** We thank the anonymous reviewers for their valuable and constructive comments.

**Conflicts of Interest:** The authors declare no conflict of interest.

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
