# Peer review of "Transport Cost Estimation Model of the Agroforestry Biomass in a Small-Scale Energy Chain"

_forests, doi:10.3390/f12020158_

Round 1

Reviewer 1 Report

Generally, There are already quite a few studies that address similar research in specific conditions. Therefore, I am surprised by the relatively small number of referenced works in the introduction. Also, new knowledge and approaches that result from the work do not necessarily mean increased interest from readers of the journal. 

Material and Methods

Line 182: In downloaded manuscript i don´t see the Equation Nr. 2.

Results:

Figure 6 it would be appropriate to enlarge the image. It is not very clear in this form.

Discussion:

It would be appropriate to support the discussion by further comparing works that address similar economic models of transport and logistics of biomass for energy purposes. By broadening the discussion, your work will also get a better international dimension and highlight the benefits of your proposed model.

Figure 8 and its description fit, in my opinion, into the results section rather than the discussion section.

Conclusion:

In conclusion, I miss the statement whether the model used is suitable only for the conditions of Italy, or it can be used in other conditions (eg road network, biomass resources, etc ..).

In my opinion, the work is only regional in nature, which may affect the interest of readers. Literary research as well as the discussion part need to be strengthened.

Author Response

RESPONSE to Reviewer 1 Comments

Point 1: Generally, there are already quite a few studies that address similar research in specific conditions. Therefore, I am surprised by the relatively small number of referenced works in the introduction. Also, new knowledge and approaches that result from the work do not necessarily mean increased interest from readers of the journal. 

Response 1: as suggested, the introduction was improved (line 40-43 and 51-61), and further studies were cited (line 42,46,52,60,61).

Point 2: Line 182: In downloaded manuscript I don´t see the Equation Nr. 2.

Response 2: Equation Nr. 2 has been inserted.

Point 3: Figure 6 it would be appropriate to enlarge the image. It is not very clear in this form.

Response 3: A larger image size has been restored.

Point 4: It would be appropriate to support the discussion by further comparing works that address similar economic models of transport and logistics of biomass for energy purposes. By broadening the discussion, your work will also get a better international dimension and highlight the benefits of your proposed model.

Response 4: the discussion section has been improved (line 295-302 and 325-327) and new references of studies on the transport and logistics of biomass for energy purposes have been added.

Point 5: Figure 8 and its description fit, in my opinion, into the results section rather than the discussion section.

Response 5: Figure 8 and its comments have been moved to the results section (line 263-273).

Point 6: In conclusion, I miss the statement whether the model used is suitable only for the conditions of Italy, or it can be used in other conditions (eg road network, biomass resources, etc.).

Response 6: the conclusions has been improved and further specification has been added in support of the proposed model (line 330-336).

Point 7: In my opinion, the work is only regional in nature, which may affect the interest of readers. Literary research as well as the discussion part need to be strengthened.

Response 7: see also response 4 and 6, the literary research and discussion section has been implemented.

Reviewer 2 Report

According to actors , the wood residues transport from its harvest to further production place can be the one of the most economically expensive operations in the entire process of producing energy from biomass. The article describes transport costs calculation by the geographic identification of different biomass sources in the supply chain.

Basically, the article refers the use of logging residues from selected areas in Italy. In the introduction (line 42 - 43), authors describes the ways of logging residues management was mentioned. In this place it is worth to mention a few alternative ways of wider logging residues use. I propose to discuss this with reference to the articles:

Nurek, T., Gendek, A., & Roman, K. (2018). Forest Residues as a Renewable Source of Energy: Elemental Composition and Physical Properties. BioResources, 14(1), 6-20. Retrieved from https://ojs.cnr.ncsu.edu/index.php/BioRes/article/view/BioRes_14_1_6_Nurek_Forest_Residues_Renewable_Energy/6480

Tomasz Nurek, Arkadiusz Gendek, Kamil Roman, Magdalena Dąbrowska, 2019, The effect of temperature and moisture on the chosen parameters of briquettes made of shredded logging residues, Biomass and Bioenergy, Volume 130, 105368, ISSN 0961-9534, https://doi.org/10.1016/j.biombioe.2019.105368.

https://www.researchgate.net/publication/336069550_The_effect_of_temperature_and_moisture_on_the_chosen_parameters_of_briquettes_made_of_shredded_logging_residues

These above sources broadly describe the research and management of logging residues

line 108 - equation is missing

line 121-122 - how many times the trimming cycle occurs it was not taken into account.

line 114 - 130 - Generally, these data require reconstruction (e.g. tabular) and standardization of units, because sometimes is: Mg or kg * ha-1, and sometimes: kg * tree-1. The reader has to do some extra counts.

line 231-245 - Figure 4 and Figure 5 are the same

line 244-245 - wrong figure description - economic sustainability (b)

Author Response

RESPONSE to Reviewer 2 Comments

Point 1: According to actors, the wood residues transport from its harvest to further production place can be the one of the most economically expensive operations in the entire process of producing energy from biomass. The article describes transport costs calculation by the geographic identification of different biomass sources in the supply chain.

Point 2: Basically, the article refers the use of logging residues from selected areas in Italy. In the introduction (line 42 - 43), authors describes the ways of logging residues management was mentioned. In this place it is worth to mention a few alternative ways of wider logging residues use. I propose to discuss this with reference to the articles:

Nurek, T., Gendek, A., & Roman, K. (2018). Forest Residues as a Renewable Source of Energy: Elemental Composition and Physical Properties. BioResources, 14(1), 6-20. Retrieved from https://ojs.cnr.ncsu.edu/index.php/BioRes/article/view/BioRes_14_1_6_Nurek_Forest_Residues_Renewable_Energy/6480

Tomasz Nurek, Arkadiusz Gendek, Kamil Roman, Magdalena Dąbrowska, 2019, The effect of temperature and moisture on the chosen parameters of briquettes made of shredded logging residues, Biomass and Bioenergy, Volume 130, 105368, ISSN 0961-9534, https://doi.org/10.1016/j.biombioe.2019.105368.

https://www.researchgate.net/publication/336069550_The_effect_of_temperature_and_moisture_on_the_chosen_parameters_of_briquettes_made_of_shredded_logging_residues

These above sources broadly describe the research and management of logging residues

Response 2: the suggestion was accepted, the discussion on this point was expanded and further references added (line 58-61).

Point 3: line 108 – equation is missing

Response 3: Equation 2 (line 182 instead 108) has been added.

Point 4: line 121-122 - how many times the trimming cycle occurs it was not taken into account.

Response 4: added in the text "with annual pruning"

Point 5: line 114 - 130 - Generally, these data require reconstruction (e.g. tabular) and standardization of units, because sometimes is: Mg or kg * ha-1, and sometimes: kg * tree-1. The reader has to do some extra counts.

Response 5: A new table (Table 2) (line 119) has been inserted to replace the descriptive part. The units of measurement have been standardized as suggested. A new Equation 3 has been added to improve this part of the methodology section.

Point 6: line 231-245 - Figure 4 and Figure 5 are the same

Response 6: Restored the correct position of Figure 4 and Figure 5

Point 7: line 244-245 - wrong figure description - economic sustainability (b)

Response 7: the description of the figure is correct: figure 5a refers to the cost of transport; Figure 5b refers to economic sustainability.

Round 2

Reviewer 1 Report

Thank you for the improvements on the manuscript file according to my suggestions.

Reviewer 2 Report

The manuscript has been revised in line with the comments made. Everything is well prepared and in my opinion the manuscript is suitable for publication.